# Efficient Out-of-Distribution Detection via CVAE data Generation

## Abstract

Recently, contrastive loss with data augmentation and pseudo class creation has been shown to produce markedly better results for out-of-distribution (OOD) detection than previous methods. However, a major shortcoming of this approach is that it is extremely slow due to significant increase in the data size and the number of classes and the quadratic complexity of pairwise similarity computation. This paper proposes a novel and simple method that can build an effective data generator using Conditional Variational Auto-Encoder (CVAE) to generate pseudo OOD samples. Based on the generated pseudo OOD data, a flexible and efficient OOD detection method is proposed through fine-tuning, which achieves results comparable to the state-of-the-art OOD detection techniques, but the execution speed is at least 10 times faster. Also importantly, the proposed approach is in fact a general framework that can be applied to many existing OOD methods and improve them via the proposed fine-tuning. We have combined it with the best baseline OOD models in our experiments to produce new state-of-the-art results.

## 1 Introduction

With the development of deep learning, a well-trained neural network model is able to obtain very high accuracy on its testing data. However, when exposed to samples or data instances drawn from a distribution that is far from the training distribution (called *In-distribution* (IND)), the model may make arbitrary predictions under the known framework (Nguyen et al., 2015; Recht et al., 2019). This limits the model's application in a broad range of applications, including secure authentication (Sharif et al., 2016), autonomous driving (Nitsch et al., 2020) and medical diagnosis (Caruana et al., 2015) as in these applications novel or *out-of distribution* (OOD) data instances occur frequently. Therefore, OOD detection (Hodge & Austin, 2004), which aims to detect abnormal or novel data that are very different from the training data, is an important research task.

Many approaches have been proposed to solve this problem, from distance-based methods (Bendale & Boult, 2015; 2016; Gunther et al., 2017; Júnior et al., 2017), to generative models (Ge et al., 2017; Neal et al., 2018; Oza & Patel, 2019; Nalisnick et al., 2018) and self-supervised learning (Bergman & Hoshen, 2020; Golan & El-Yaniv, 2018; Hendrycks et al., 2019). Recently, contrastive learning has been shown highly effective in many applications (Hjelm et al., 2018; Oord et al., 2018; Chen et al., 2020b;a; Falcon & Cho, 2020). Applying supervised contrastive learning and data augmentation, the recent CSI system has produced state-of-the-art (SOTA) OOD detection results (Tack et al., 2020).

However, data augmentation-based contrastive learning also has some drawbacks. First, designing data augmentation functions and deciding how to use various types of augmented data in contrastive learning involve a great deal of trial and error and manual work. That is, there is a large number of transformations (or augmentations) such as cropping, rotation and gray-scaling that can be exerted on images but not all of them may benefit the end tasks. In SimCLR (Chen et al., 2020a), systematic experiments have been reported to explore the augmentations' influence on classification tasks. Based on SimCLR's results, CSI (Tack et al., 2020) chooses several operations for OOD detection.

Second, contrastive learning with data augmentation is very time-consuming to run and resource-hungry due to a large amount of augmented data and quadratic pairwise similarity computation during training. For example, CSI creates 3 shifted instances for each original image sample and the 4 images are then subjected to an additional augmentation operation. Each image is finally expanded to 8 images or samples. Furthermore, every 2 samples in the augmented batch is treated

as a pair to calculate contrastive loss. The performance is negatively affected if the batch size is not large enough. Therefore, it is necessary to use a large batch size, which demands a huge amount of memory and takes a very long time to run. It is thus not suitable for applications on hardware devices that do not have the required resources such as edge devices.

In this paper, we propose a general and much more efficient solution, called CGA (*CVAE-based Generative data Augmentation for OOD detection*). CGA consists of two parts. The first part generates pseudo OOD data and the second part makes use of the pseudo OOD data to train an OOD detection model. We discuss the first part first. OOD detection is basically a classification problem but the challenge is that there is no OOD data to use in training. This paper proposes a novel method to generate pseudo OOD data. This method works in the latent space of a Conditional Variational Auto-Encoder (CVAE) and uses CVAE's decoder to generate pseudo OOD data. CVAE is able to generate instances from the training distribution on the basis of latent representations consisting of conditional information and variables sampled from a prior distribution of CVAE, normally the Gaussian distribution. If the latent space features or representations are created with some *abnormal conditional information*, the CVAE will generate "bad" instances but such instances can serve as effective pseudo OOD data. The second part of CGA is a fine-tuning framework that can make use of the generated pseudo OOD data to fine-tune any classification model built using only the IND data. Applying the framework to a simple IND classification model, we can already produce comparable results to existing SOTA contrastive learning models but much more efficient. Applying it to the existing SOTA methods, we can improve their results too.

Our contributions can be summarized as follows: **(1)** We propose to employ a CVAE structure to generate pseudo OOD samples by providing some synthetic conditional information, which, to our knowledge, has not been done before. **(2)** We design a two-stage framework to train an OOD detection model by leveraging the generated pseudo OOD data. The first stage simply builds a normal classification model using only the IND data. The second stage fine-tunes the model using the generated OOD data to produce an effective OOD detection model. Extensive experiments show that this approach achieves comparable performance to the state-of-the-art (SOTA) contrastive learning methods in OOD detection, while consuming only one-tenth of the execution time. **(3)** Equally importantly, the proposed framework can be applied to existing SOTA OOD detection models to improve them to produce new SOTA results.

## 2 RELATED WORK

**Out-of-distribution Detection.** It is well-known that the discriminative neural networks can produce overconfident predictions on out-of-distribution (OOD) inputs (Hendrycks & Gimpel, 2016; Lakshminarayanan et al., 2016). The early idea to solve this problem focuses on modifying softmax scores to obtain calibrated confidence for OOD detection (Bendale & Boult, 2016; Guo et al., 2017). In addition, many other score functions have been proposed, such as likelihood ratio (Ren et al., 2019), input complexity (Serrà et al., 2019) and typicality (Nalisnick et al., 2019). A recent work utilizes Gram matrices to characterize activity patterns and identify anomalies in Gram matrix values to do OOD detection (Sastry & Oore, 2020). Some methods found that auxiliary anomalous data significantly improve detection performance (Hendrycks et al., 2018; Mohseni et al., 2020) and thus generative models are adopted to anticipate the distribution of novel samples. In some of these methods, generated data are treated as OOD samples to optimize the decision boundary and calibrate the confidence (Ge et al., 2017; Vernekar et al., 2019). In some other works, generative models such as auto-encoders (Zong et al., 2018; Pidhorskyi et al., 2018) and generative adversarial networks (GAN) (Deecke et al., 2018; Perera et al., 2019) are used to reconstruct the training data. During the training of a GAN model, low quality samples acquired by the generator can also work as OOD data directly (Pourreza et al., 2021). Owing to the fact that the model can hardly be generalized to unknown data, the reconstruction loss can help detect OOD data. There are also works using auxiliary OOD data to fine-tune the model (Liu et al., 2020). Fort et al. (2021) showed that using pre-trained representations and taking few-shot outlier exposure can improve the results. Recently, self-supervised techniques have been applied to OOD detection. It focuses on acquiring rich representations through training with some pre-defined tasks (Gidaris et al., 2018; Kolesnikov et al., 2019). Self-supervised models show outstanding performance on OOD detection tasks (Kolesnikov et al., 2019; Bergman & Hoshen, 2020). CSI (Tack et al., 2020) is a representative method (see more below), which uses contrastive learning and data augmentation to improve the features of all labeled

IND data and produce state-of-the-art (SOTA) results. Some researchers also tried to improve contrastive learning based methods (Sehwag et al., 2021) and proposed distance-based methods (Miller et al., 2021). However, based on our experiments, CSI outperforms them. Our method falls into the generative approach. Unlike existing methods that use perturbations to anticipate OOD data, our method uses synthetic conditions and CVAE to obtain effective and diverse pseudo OOD data.

**Contrastive Learning**. Contrastive learning learns representations by contrasting positive pairs against negative pairs (Hadsell et al., 2006). It has been applied to various domains (Oord et al., 2018). Recently, a new method called SimCLR (Chen et al., 2020a) was proposed to create sample pairs via data augmentation. It is effective but also very time and resource consuming. SimCLR also shows that contrastive learning benefits more from larger batch sizes and longer training time. CSI (Tack et al., 2020) proposes that augmentation can not only help construct positive pairs but also negative pairs and makes use of them to detect OOD samples with supervised contrastive learning. It obtains the SOTA OOD detection results with labeled IND data. However, due to contrastive learning, it is extremely slow and memory demanding. Our proposed method generates pseudo OOD using CVAE, avoiding the use of contrastive loss, and is much more efficient than CSI.

**Auto-Encoder** Auto-Encoder (AE) is a family of unsupervised neural networks (Rumelhart et al., 1986; Baldi & Hornik, 1989). A basic AE consists of an encoder and a decoder. The encoder encodes the input data into a low-dimensional hidden representation and the decoder transforms the representation back to the reconstructed input data (Vincent et al., 2008; Chen et al., 2012; Hinton et al., 2006). Variational auto-encoder is a special kind of AE (Kingma & Welling, 2013). It encodes the input as a given probability distribution (usually Gaussian) and the decoder reconstructs data instances according to variables sampled from that distribution. CVAE is an extension of VAE (Kingma et al., 2014). It encodes the label or conditional information into the latent representation so that a CVAE can generate new samples from specified class labels. CVAE makes it possible to control the generating process, i.e., to generate samples with features of specified classes. We make use of this property of CVAE to generate high quality pseudo OOD data.

## 3 PROPOSED CGA METHOD

In tasks related to out-of-distribution (OOD) detection, the problem of recognition is commonly formulated as a classification problem. The main challenge is that an important class, OOD data, is not available. Therefore, to effectively train an OOD detection model, an intuitive idea is to generate pseudo OOD data and use them together with the IND data to train the model. As we mentioned earlier, data augmentation and contrastive learning have been shown especially effective for this purpose. However, this approach is extremely inefficient. We propose to use Conditional Variational Auto-Encoder (CVAE) to generate pseudo OOD data and present a new fine-tuning framework to leverage the generated pseudo OOD data to train an OOD detection model.

### 3.1 CONDITIONAL VARIATIONAL AUTO-ENCODER

Conditional Variational Auto-Encoder (CVAE) is derived from Variational auto-encoder (VAE). We first introduce VAE which is a conditional directed graphical model consisting of three main parts, an encoder $q_\phi(\cdot)$ with parameters $\phi$, a decoder $p_\theta(\cdot)$ with parameters $\theta$ and a loss function $\mathcal{L}(\mathbf{x}; \theta, \phi)$, where $\mathbf{x}$ represents an input sample. The loss function is as follows:

$$\mathcal{L}(\mathbf{x}; \theta, \phi) = -\mathbb{E}_{\mathbf{z} \sim q_\phi(\mathbf{z}|\mathbf{x})}[\log p_\theta(\mathbf{x}|\mathbf{z})] + KL(q_\phi(\mathbf{z}|\mathbf{x})||p_\theta(\mathbf{z})) \tag{1}$$

where $q_\phi(\mathbf{z}|\mathbf{x})$ is a proposal distribution to approximate the prior distribution $p_\theta(\mathbf{z})$, $p_\theta(\mathbf{x}|\mathbf{z})$ is the likelihood of the input $\mathbf{x}$ with a given latent representation $\mathbf{z}$, and $KL(\cdot)$ is the function to calculate Kullback-Leibler divergence. $q_\phi(\mathbf{z}|\mathbf{x})$ is the encoder and $p_\theta(\mathbf{x}|\mathbf{z})$ is the decoder. In Eq.(1), the expected negative log-likelihood term encourages the decoder to learn to reconstruct the data with samples from the latent distribution. The KL-divergence term forces the latent distribution to conform to a specific prior distribution such as the Gaussian distribution. After training, a VAE can generate data using the decoder $p_\theta(\mathbf{x}|\mathbf{z})$ with a set of latent variables $\mathbf{z}$ sampled from the prior distribution $p_\theta(\mathbf{z})$. Commonly, the prior distribution is the centered isotropic multivariate Gaussian $p_\theta(\mathbf{z}) = \mathcal{N}(\mathbf{z}; \mathbf{0}, \mathbf{I})$.

However, VAE does not consider the class label information which is available in classification datasets and thus has difficulty generating data of a particular class. Conditional variational Auto-

Encoder (CVAE) was introduced to extend VAE to address this problem. It improves the generative process by adding a conditional input information into latent variables so that a CVAE can generate samples with some specific characteristics or from certain classes. We use $c$ to denote the prior class information. The loss function for CVAE can be written as follows:

$$\mathcal{L}(\mathbf{x}; \theta, \phi) = -\mathbb{E}_{\mathbf{z} \sim q_\phi(\mathbf{z}|\mathbf{x})}[\log p_\theta(\mathbf{x}|\mathbf{z}, c)] + KL(q_\phi(\mathbf{z}|\mathbf{x}, c)||p_\theta(\mathbf{z}|c)) \tag{2}$$

One implementation of CVAE uses a one-hot vector to represent a class label $y_c$, and a weight matrix is multiplied to it to turn the one-hot vector to a class embedding $\mathbf{y}_c$. Then a variable $\mathbf{z}$, generated from the prior distribution $p_\theta(\mathbf{z})$, is concatenated with $\mathbf{y}_c$ to construct the whole latent variable. Finally, the generated instance $p_\theta(\mathbf{x}|\mathbf{z}, c)$ of class $c$ is produced. We can formulate the process as:

$$p_\theta(\mathbf{x}|\mathbf{z}, c) = p_\theta(\mathbf{x}|[\mathbf{y}_c, \mathbf{z}]) \tag{3}$$

### 3.2 GENERATING PSEUDO OOD DATA

The ability of CVAE to control the generating process using the conditional information (e.g. class label in our case) inspired us to design a method to generate possible OOD samples. This is done by the conditional decoder using atypical prior information $c$ in $p_\theta(\mathbf{x}|\mathbf{z}, c)$. As introduced before, OOD data need to be different from in-distribution data but also resemble them. The continuity property of CVAE, which means that two close points in the latent space should not give two completely different contents once decoded (Cemgil et al., 2019), ensures that we can manipulate CVAE latent space features to generate high quality pseudo OOD data. Since we have no information of the future OOD data, we have to make use of the existing training data (i.e., in-distribution data) to construct pseudo OOD data. We can provide it with pseudo label information to generate pseudo OOD data.

Specifically, we propose to construct pseudo class embedding by combining the embeddings of two existing classes in the in-distribution training data. The formulation is as follows:

$$p_\theta(\mathbf{x}|\mathbf{z}, \mathbf{k}, c_i, c_j) = p_\theta(\mathbf{x}|[\mathbf{k} * \mathbf{y}_{c_i} + (1 - \mathbf{k}) * \mathbf{y}_{c_j}, \mathbf{z}]) \tag{4}$$

where $\mathbf{k}$ is a vector generated from Bernoulli distribution $\mathcal{B}(0.5)$ with the same length as the class label embedding. $\mathbf{k}$ is basically for the system to randomly select the vector components of the two class embeddings with equal probability. Such a generated sample $p_\theta(\mathbf{x}|\mathbf{z}, \mathbf{k}, c_i, c_j)$ will not likely to be an instance of either class $c_i$ or $c_j$ but still keep some of their characteristics, which meets the need for pseudo OOD data. Furthermore, the pseudo class embedding has a great variety, owing to the diverse choices of classes and the vector $\mathbf{k}$. To generate pseudo OOD samples, we also need to sample $\mathbf{z}$ from the encoder. In the training of CVAE, we ensure that $\mathbf{z}$ fits the Gaussian distribution $\mathcal{N}(\mathbf{0}, \mathbf{I})$. To sample $\mathbf{z}$, we use another flatter Gaussian distribution $\mathcal{N}(\mathbf{0}, \sigma^2 * \mathbf{I})$, where $\sigma > 1 \in \mathbb{Z}$, to make the generated samples highly diverse.

### 3.3 TRAINING PROCESS FOR OOD DETECTION

With the generated pseudo OOD samples and the original in-distribution (IND) training data, training an OOD detection model consists of two stages.

**Stage 1 (IND classifier building and CVAE training):** Only the original IND data is used to train a classification model $\mathcal{C}$. The classification model can be decomposed into two functions $f$ and $h$, where $f$ is the final linear *classifier* and $h$ is the *feature extractor*. $f(h(\mathbf{x}))$ is the classification output. A separate CVAE model is also trained for generating pseudo OOD data.

**Stage 2 (fine-tuning with pseudo OOD data):** We keep the trained feature extractor $h$ fixed (or frozen) and fine-tune only the classification layer $f$ for OOD detection.

The proposed CGA approach is in fact a framework, which is illustrated in Figure 1 together with the two-stage training process. The framework is very flexible as the classifier in the first stage can use any model. Stage 2 is also flexible and can use many approaches. Here we introduce two specific methods. They produce results comparable to the state-of-the-art OOD detection models, but are very simple and very efficient. In fact, as we will see in the experiments, fine-tuning the existing state-of-the-art (SOTA) OOD detection models can also improve them to produce new SOTA results.

**CGA-softmax.** In the fine-tuning stage, we simply add additional class (let us call it the OOD class) in the classification layer to accept the pseudo OOD data. If the IND data has $N$ classes,

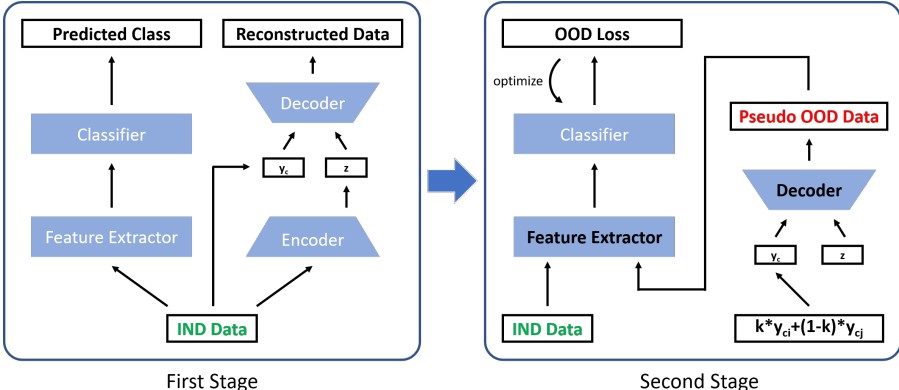

First Stage                    Second Stage

Figure 1: The CGA framework and its training process. The OOD loss can be the cross entropy loss in CGA-softmax, cross-entropy+energy loss in CGA-energy, or other possible losses.

we add parameters to the classifier to make it output $N + 1$ logits. These added parameters related to $(N + 1)$th OOD class is randomly initialized. We then train the model by only fine-tuning the classification layer using the cross entropy loss with feature extractor trained in stage 1 fixed. Finally, we use the softmax score of the $(N + 1)$th class as the OOD score.

**CGA-energy.** In this approach, we add an energy loss to the cross entropy loss ($\mathcal{L}_{ent} + \lambda \mathcal{L}_{energy}$) to fine-tune the classification layer using the IND data and the pseudo OOD data to produce an OOD score for each test instance. In this case, no OOD class is added. The energy loss is as follows,

$$\mathcal{L}_{energy} = \mathbb{E}_{\mathbf{x}_{in} \sim \mathcal{D}_{in}} (\max(0, E(\mathbf{x}_{in}) - m_{in}))^2 + \mathbb{E}_{\mathbf{x}_{out} \sim \mathcal{D}_{out}} (\max(0, m_{out} - E(\mathbf{x}_{out})))^2 \quad (5)$$

where $\mathcal{D}_{in}$ denotes the IND training data, $\mathcal{D}_{out}$ denotes generated pseudo OOD data, and $m_{in}$ and $m_{out}$ are margin hyper-parameters. The idea of this loss is to make the OOD data get similar values for all $N$ logits so that they will not be favored by any $N$ IND data classes. Here $N$ is the number of classes of the IND data. As the loss function shows, the OOD data is necessary. This loss was used in (Liu et al., 2020), which has to employ OOD data but such OOD data is often not available in practice. This loss cannot be used by other OOD methods since they have no OOD data available (Bendale & Boult, 2016; Khosla et al., 2020; Tack et al., 2020). However, this is not an issue for us as we have pseudo data to replace real OOD training data.

Stage 2 produces an energy score calculated from a classification model for OOD detection:

$$E(\mathbf{x}; f(h)) = -T \cdot \log \sum_{i=1}^{N} e^{f_i(h(\mathbf{x}))/T} \quad (6)$$

where $E(\mathbf{x}; f(h(\cdot)))$ denotes the energy of instance $\mathbf{x}$ with the classification model $f(h(\cdot))$, which maps $\mathbf{x}$ to $N$ logits, where $N$ is the number of classes in the IND data, $f_i(h(\mathbf{x}))$ is the $i$-th logit and $T$ is the temperature parameter.

## 4   EXPERIMENTS

We construct OOD detection tasks using benchmark datasets and compare our proposed technique CGA with the state-of-the-art existing methods. *The code of CGA has been submitted.*

### 4.1   EXPERIMENT SETTINGS AND DATA PREPARATION

We use two experimental setups to evaluate our system.

**Setting 1 - OOD Detection on the Same Dataset:** In this setting, IND (in-distribution) and OOD instances are from different classes of the same dataset. This setting is often called *open-set detection*. We use the following 4 popular datasets for our experiments in this setting.

(1) **MNIST** (LeCun et al., 2010): A handwritten digit classification dataset of 10 classes. The dataset has 70,000 examples/instances, with the splitting of 60,000 for training and 10,000 for testing.

(2) **CIFAR-10** (Krizhevsky & Hinton, 2010): An image classification dataset consisting of 60,000 32x32 color images of 10 classes with the splitting of 50,000 for training and 10,000 for testing.

(3) **SVHN** (Netzer et al., 2011): A colorful street view house number classification dataset of 10 classes. It contains 99289 instances with the splitting of 73257 for training and 26032 for testing.

(4) **TinyImageNet** (Le & Yang, 2015): A classification dataset of 200 classes. Each class contains 500 training samples and 50 testing samples. The resolution of the images is 64x64.

We follow the data processing method in (Sun et al., 2020; Zhou et al., 2021) to split known and unknown classes. For each dataset, we conduct 5 experiments using different splits of known (IND) and unknown (OOD) classes. These same 5 splits are used by all baselines and our system. Following (Sun et al., 2020), for MNIST, CIFAR-10 and SVHN, 6 classes are chosen as IND classes, and the other 4 classes are regarded as OOD classes. The following 5 fixed sets of IND classes, 0-5, 1-6, 2-7, 3-8, and 4-9, are used and they are called **partition 1, 2, 3, 4**, and **5**, respectively. The rest 4 classes in each case serve as the OOD classes. For TinyImageNet, each set of IND data contains 20 classes and the sets of IND classes in the 5 experiments are 0-19, 40-59, 80-99, 120-139, and 160-189 respectively. The rest 180 classes are regarded as the OOD classes.

**Setting 2 - OOD Detection on Different Datasets:** The IND data and OOD data come from different datasets. Following (Tack et al., 2020), we use CIFAR-10 as the IND dataset and each of the following datasets as the OOD dataset.

(1) **SVHN** (Netzer et al., 2011): See above. All 26032 testing samples are used as OOD data.

(2) **LSUN** (Yu et al., 2015): This is a large-scale scene understanding dataset with a testing set of 10,000 images from 10 different scenes. Images are resized to 32x32 in our experiment.

(3) **LSUN-FIX** (Tack et al., 2020): To avoid artificial noises brought by general resizing operation, this dataset is generated by using a fixed resizing operation on LSUN to change the images to 32x32.

(4) **TinyImageNet** (Le & Yang, 2015): See above. All 10000 testing samples are used as OOD data.

(5) **ImageNet-FIX** (Le & Yang, 2015): 10,000 images are randomly selected from the training set of ImageNet-30, excluding "airliner", "ambulance", "parkingmeter", and "schooner" classes to avoid overlapping with CIFAR-10. A fixed resizing operation is applied to transform the images to 32x32.

(6) **CIFAR100** (Krizhevsky et al., 2009): An image classification dataset consisting of 60,000 32x32 color images of 100 classes. Its 10,000 test samples are used as the OOD data.

## 4.2 BASELINES

We compare with the following state-of-the-art baselines using the official code from their authors.

(1) **Softmax**: This is the popular classification score model. The highest softmax probability is used as the confidence score for OOD detection.

(2) **OpenMax** (Bendale & Boult, 2016): This method combines the softmax score with the distance between the test sample and IND class centers to detect OOD data.

(3) **ODIN** (Liang et al., 2017): This method improves the OOD detection performance of a pre-trained neural network by using temperature scaling and adding small perturbations to the input.

(4) **Maha** (Lee et al., 2018): This method uses Mahalanobis distance to evaluate the probability that an instance belongs to OOD.

(5) **CCC** (Lee et al., 2017): This is a GAN-based method, jointly training the classification and pseudo OOD generator for OOD detection.

(6) **OSRCI** (Neal et al., 2018): This method also uses GAN to generate pseudo instances and further improves the model to predict novelty (OOD) examples.

(7) **CAC** (Miller et al., 2021): This is a distance-based method, using the Class Anchor Clustering loss to cluster IND samples tightly around the anchored centres for OOD detection.

(8) **SupCLR** (Khosla et al., 2020): This is a contrastive learning based method. It extends contrastive learning to fully-supervised setting to improve the quality of features for classification.

(9) **CSI** (Tack et al., 2020): This is also a supervised contrastive learning method. It uses extensive data augmentations to generate shifted data instances. It also has a score function that benefits from the augmented instances for OOD detection.

For Softmax, OpenMax and OSRCI, we use OSRCI's implementation[1]. For SupCLR and CSI, we use CSI's code[2]. For ODIN, Maha, CCC and CAC, we use their original code[3456]. We also use their default hyper-parameters.

### 4.3 IMPLEMENTATION DETAILS

For MNIST, we use a 9-layer CNN as the encoder (feature extractor) and a 2-layer MLP as the projection head. CVAE includes a 2-layer CNN as the encoder and a 2-layer deconvolution network (Zeiler et al., 2011) as the decoder, as well as two 1-layer MLPs to turn features into means and variations. For the other datasets, the encoder is a ResNet18 (He et al., 2016) and the projection head is a 2-layer MLP. CVAE also uses ResNet18 as the encoder, and 2 residual blocks and a 3-layer deconvolution network as the decoder. The mean and variation projection are completed by two 1-layer MLPs. During the first stage of training, we use Adam optimizer (Kingma & Ba, 2014) with $\beta_1 = 0.9$, $\beta_2 = 0.999$ and learning rate of 0.001. We train both the classification model and CVAE model for 200 epochs with batch size 512. In the second stage, the learning rate is set to 0.0001 and the fine-tuning process with the generated pseudo data are run for 10 epochs. The number of generated pseudo OOD data is the same as the IND data (we will study this further shortly). Each batch has 128 IND samples and 128 generated OOD samples. There is no special hyper-parameter for CGA-softmax in stage 2. For CGA-energy, two special hyper-parameters of the energy loss $m_{in}$ and $m_{out}$ are decided at the beginning of stage 2 by IND and pseudo data. We calculate the energy of all training IND data and generated pseudo data. Then $m_{in}$ and $m_{out}$ are chosen to make 80% of IND data's energy larger than $m_{in}$ and 80% of pseudo data's energy smaller than $m_{out}$. This ensures that 80% of data get non-zero loss. We use the same 1 NVIDIA-GeForce-RTX-2080Ti GPU for the experiments of evaluating the running speed of different methods.

### 4.4 RESULTS AND DISCUSSIONS

Table 1 shows the results of the two OOD detection settings on different datasets. Due to the large image size, a large number of IND classes and a large batch size requirement, we were unable to run SupCLR and CSI using TinyImageNet on our hardware and thus do not have their results in Setting 1. On average, our CGA model can achieve the best results in Setting 1. In Setting 2, benefiting from strong features learned using contrastive loss, CSI performs the best on average and our CGA-e (CGA-energy) is slightly weaker than CSI. But from Table 3 we learn that CGA is much more efficient than the contrastive learning methods. With comparable overall performances on OOD detection, CGA spends only about 10% of CSI's training time. We also notice that our CGA-s (CGA-softmax) is slightly weaker than CGA-e, which shows that the energy function is effective.

Equally importantly, Table 2 demonstrates that CGA's fine-tuning (stage 2) can improve the 3 best performing baselines in Table 1, i.e., GAN-based OSRCI and contrastive learning based SupCLR and CSI. Here after each baseline finishes its training, we apply fine-tuning of CGA's stage 2 to fine-tune the trained model using CGA-energy. Although CSI already produces the best OOD detection result, it is improved further by our CGA framework to produce a new SOTA result.

### 4.5 ABLATION STUDY

We now perform the ablation study with various options of our system and report the AUC scores on 5 partitions of the CIFAR10 dataset in Setting 1 - OOD detection on the same dataset.

---

[1]https://github.com/lwneal/counterfactual-open-set
[2]https://github.com/alinlab/CSI
[3]https://github.com/facebookresearch/odin
[4]https://github.com/pokaxpoka/deep_Mahalanobis_detector
[5]https://github.com/alinlab/Confident_classifier
[6]https://github.com/dimitymiller/cac-openset

Table 1: AUC (Area Under the ROC curve) (%) on detecting IND and OOD samples on 2 settings. Results are averaged over the 5 partitions on setting 1. CGA-s is the CGA-softmax model and CGA-e is the CGA-energy model. Every experiment was run 5 times.

| Datasets | Softmax | OpenMax | ODIN | Maha | CCC | OSRCI | CAC | SupCLR | CSI | CGA-s | CGA-e |
|---|---|---|---|---|---|---|---|---|---|---|---|
| **Setting 1 - OOD Detection on the Same Dataset** | | | | | | | | | | | |
| MNIST | 97.6±0.7 | 98.1±0.5 | 98.1±1.1 | 98.4±0.4 | 94.2±0.8 | 98.3±0.9 | **99.2±0.1** | 97.1±0.2 | 97.2±0.3 | 98.3±0.2 | 99.0±0.2 |
| CIFAR-10 | 65.5±0.5 | 66.9±0.4 | 79.4±1.6 | 73.4±2.2 | 74.0±1.4 | 67.5±0.8 | 75.9±0.7 | 80.0±0.5 | 84.7±0.3 | **86.3±1.2** | 85.6±0.6 |
| SVHN | 90.3±0.5 | 90.7±0.4 | 89.4±2.0 | 91.5±0.6 | 64.6±2.3 | 91.7±0.2 | 93.8±0.2 | 93.8±0.2 | **93.9±0.1** | 91.8±0.5 | 92.1±0.4 |
| TinyImageNet | 57.5±0.7 | 57.9±0.2 | 70.9±1.5 | 56.3±1.9 | 51.0±1.2 | 58.1±0.4 | 71.9±0.7 | \ | \ | 72.2±0.5 | **73.7±0.6** |
| Average | 77.8 | 78.4 | 84.5 | 79.9 | 71.0 | 78.9 | 85.2 | \ | \ | 87.2 | **87.6** |
| **Setting 2 - OOD Detection on Different Datasets (CIFAR-10 as IND)** | | | | | | | | | | | |
| SVHN | 80.2±1.8 | 82.7±1.9 | 83.2±1.5 | 97.5±1.6 | 83.3±0.8 | 80.2±1.8 | 87.3±4.6 | 97.3±0.1 | **97.9±0.1** | 95.8±0.6 | 96.2±2.5 |
| LSUN | 70.1±2.5 | 72.2±1.8 | 82.1±1.9 | 61.5±5.0 | 85.6±2.3 | 79.9±1.8 | 89.1±3.4 | 92.8±0.5 | 97.7±0.4 | 96.8±1.4 | **97.7±0.9** |
| LSUN-FIX | 76.7±0.8 | 75.6±1.2 | 84.1±1.7 | 77.8±2.1 | 86.6±1.6 | 78.2±0.5 | 85.5±0.7 | 91.6±1.5 | 93.5±0.4 | **94.1±0.9** | 93.7±0.4 |
| TinyImageNet | 62.5±3.6 | 65.2±3.1 | 68.7±2.2 | 56.8±2.1 | 83.2±1.8 | 70.0±1.7 | 86.4±4.6 | 91.4±1.2 | **97.6±0.3** | 94.8±1.6 | 95.2±2.7 |
| ImageNet-FIX | 75.9±4.6 | 75.6±0.7 | 74.8±0.6 | 79.0±3.1 | 83.7±1.1 | 78.1±0.3 | 85.6±0.3 | 90.5±0.5 | **94.0±0.1** | 89.7±0.3 | 92.9±1.2 |
| CIFAR100 | 74.6±0.5 | 75.5±0.4 | 74.5±0.8 | 61.4±0.9 | 81.9±0.5 | 77.4±0.4 | 83.9±0.2 | 88.6±0.2 | **92.2±0.1** | 87.9±0.4 | 89.3±0.4 |
| Average | 73.3 | 74.5 | 77.9 | 72.3 | 84.1 | 77.3 | 86.3 | 92.0 | **95.5** | 93.2 | 94.2 |

Table 2: AUC (Area Under the ROC curve) (%) results of the original model (denoted by **original**) and the model plus fine-tuning using CGA-energy (denoted by **+CGA-e**). Almost every +CGA-e version of the baselines outperforms the original model. Every experiment was run 5 times.

| Datasets | OSRCI | | SupCLR | | CSI | |
|---|---|---|---|---|---|---|
| | original | +CGA-e | original | +CGA-e | original | +CGA-e |
| **Setting 1 - OOD Detection on the Same Dataset** | | | | | | |
| MNIST | 98.3 ±0.9 | 99.1±0.4 | 97.1±0.2 | 98.6±0.2 | 97.2±0.3 | 99.3±0.1 |
| CIFAR-10 | 67.5±0.8 | 72.3±0.6 | 80.0±0.5 | 88.9±0.5 | 84.7±0.3 | 89.8±0.6 |
| SVHN | 91.7±0.2 | 92.1±0.1 | 93.8±0.2 | 96.5±0.3 | 93.9±0.1 | 96.7±0.2 |
| TinyImageNet | 58.1±0.4 | 59.9±0.3 | \ | \ | \ | \ |
| **Setting 2 - OOD Detection on Different Datasets (CIFAR-10 as IND)** | | | | | | |
| SVHN | 80.2±1.8 | 79.3±2.5 | 97.3±0.1 | 93.0±1.7 | 97.9±0.1 | 97.8±0.6 |
| LSUN | 79.9±1.8 | 92.1±0.6 | 92.8±0.5 | 97.7±0.6 | 97.7±0.4 | 99.2±0.1 |
| LSUN-FIX | 78.2±0.5 | 81.2±1.0 | 91.6±1.5 | 94.1±0.3 | 93.5±0.4 | 96.2±0.3 |
| TinyImageNet | 70.0±1.7 | 83.2±1.7 | 91.4±1.2 | 96.3±0.8 | 97.6±0.3 | 98.7±0.3 |
| ImageNet-FIX | 78.1±0.3 | 78.5±0.2 | 90.5±0.5 | 92.9±0.3 | 94.0±0.1 | 95.7±0.1 |
| CIFAR100 | 77.4±0.4 | 77.4±0.6 | 88.6±0.2 | 90.3±0.2 | 92.2±0.1 | 92.0±0.2 |
| Average | 77.3 | 82.0 | 92.0 | 94.1 | 95.5 | 96.6 |

Table 3: Execution time (min) of each method spent in running the whole experiment on benchmark datasets for setting 1.

| Datasets | Softmax | OpenMax | ODIN | Maha | CCC | OSRCI | CAC | SupCLR | CSI | CGA-e (ours) |
|---|---|---|---|---|---|---|---|---|---|---|
| MNIST | 6 | 6 | 71 | 54 | 133 | 49 | 13 | 1260 | 1728 | 24 |
| CIFAR-10 | 20 | 20 | 61 | 56 | 111 | 70 | 49 | 1110 | 1428 | 144 |
| SVHN | 20 | 20 | 142 | 140 | 196 | 71 | 37 | 1770 | 2471 | 249 |
| TinyImageNet | 22 | 22 | 64 | 54 | 46 | 79 | 64 | \ | \ | 131 |

**CGA Stage 2**. To verify the effect of different options of stage 2, we compare the results of CGA-e model with **(1)** *without stage 2*, i.e., we directly use energy score of stage 1 to compute AUC, **(2)** *stage 2 without using pseudo OOD data*, and **(3)** *full stage 2*. Figure 2(a) shows that without stage 2, stage 1 produces poor results. Stage 2 without the generated pseudo OOD data only improves the performance slightly. The full stage 2 with the generated pseudo data greatly improves the performance of OOD detection. The experiments prove the necessity of stage 2 and the effectiveness of the generated pseudo OOD data. We do not vary stage 1 as our contribution is in stage 2.

**Amount of Pseudo OOD Data**. We run experiments of stage 2 with different numbers of generated pseudo OOD samples to further analyze their effectiveness. Figure 2(b) demonstrates that the model can benefit from only a few pseudo samples significantly. Though the amount is only 10% of the IND data, the pseudo data can improve the results markedly, which indicates that pseudo samples are highly effective. The results are similar when pseudo samples are more than half of the IND training samples. We use the same number of pseudo samples as the IND samples in all our experiments.

**Pseudo OOD Data Distribution.** The CVAE generator is trained to make the latent variables or features conform to the Gaussian distribution $\mathcal{N}(\mathbf{0}, \mathbf{I})$ (see Section 3.2). To make pseudo data diverse and different from the training data, we sample the latent variables $\mathbf{z}$ from a pseudo data sampling distribution $\mathcal{N}(\mathbf{0}, \sigma^2 * \mathbf{I})$. We conduct experiments to study the effect of the distribution. First, we

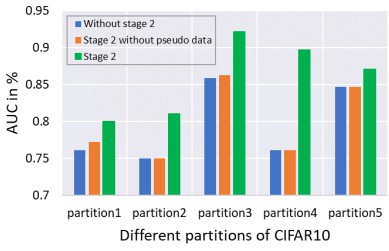 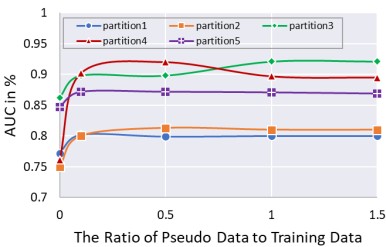

(a) Different options of CGA Stage 2    (b) Ratio of pseudo OOD data to IND data.

Figure 2: Ablation study on (a) CGA Stage 2 and (b) amount of generated pseudo OOD data.

study the influence of $\sigma$. Note that $\sigma$ is 1 in training. With larger $\sigma$ values, the sampled values will be more likely to be far from **0** (which is the mean) to make the latent features different from those seen in training. Figure 3(a) shows the results, which indicate the necessity of using $\sigma > 1$. Results are similar when $\sigma \geq 5$ and we use 5 in our experiments.

Intuitively, we may only keep latent features **z** that are far from the Gaussian distribution mean by filtering out values that are close to **0** (or the mean). We use a filtering threshold $t$ to filter out the sampled **z** whose component values are within the range $[-t, t]$. Experimental results in Figure 3(b) allow us to make the following observations. When $\sigma = 1$, as $t$ grows, the performance improves slightly. But comparing with the results in Figure 3(a), we see that a larger $\sigma$ improves the performance more. Figure 3(c) tells us that when $\sigma = 5$, the effect of filtering diminishes. For simplicity and efficiency, all our experiments employed $\sigma = 5$ without filtration.

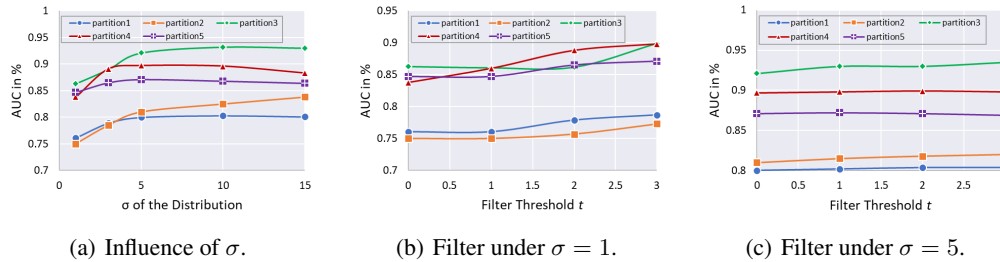

(a) Influence of $\sigma$.    (b) Filter under $\sigma = 1$.    (c) Filter under $\sigma = 5$.

Figure 3: Ablation study on different options of generating pseudo OOD data. Figure 3(a) shows AUC results of different $\sigma$ values of the sampling distribution. Figure 3(b) filters values near the center of the Gaussian distribution with $\sigma = 1$. Figure 3(c) filters values near the center of the Gaussian Distribution with $\sigma = 5$.

## 5 CONCLUSION

It has been shown recently that contrastive learning with extensive data augmentations can produce the state-of-the-art out-of-distribution detection results. Our experiments also confirmed that. However, such algorithms are extremely inefficient and resource hungry due to a large amount of augmented data and quadratic pairwise similarity computation, which makes such algorithms unsuitable for applications that do not have the required computing resources, e.g., edge devices. In this paper, we proposed a simple alternative based on data generation using CVAE which delivers similar accuracy results but is much more efficient. What is also interesting is that our proposed approach can improve the results of those state-of-the-art contrastive learning based methods too. Thus, our future work will focus on improving the detection accuracy while maintaining or further reducing the computing resources requirements. We also plan to explore pre-trained feature extractors and more advanced variants of CVAE to improve our method.

On broader impacts, we believe that the proposed approach has a great potential and may also be applicable to one-class learning, positive and unlabeled (PU) learning, and continual learning. for these problems, the ability to generate pseudo OOD data is useful.

ETHICS STATEMENT

We believe that our work has no ethic issues as we propose a general algorithm for out-of-distribution detection, which is not for any specific application. Our experimental datasets are all public domain datasets.

REPRODUCIBILITY STATEMENT

We have submitted our code in the supplementary material.

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

## A  IMPLEMENTATION DETAILS OF BASELINES

For OOD detection on the same dataset, since the partitions of in-distribution (IND) and out-of-distribution (OOD) classes from a dataset affect the results significantly, we show the detailed results of individual partitions in the next section. As for OOD detection on different datasets, we copy SupCLR and CSI's results from the CSI paper and produce our own results for the other baselines.

Table 4: AUC (Area Under the ROC curve) (%) on detecting OOD samples from the same dataset. Results of the 5 partitions of each dataset are listed in the table.

| Partitions | Softmax | OpenMax | OSRCI | SupCLR | CSI | CGA-s | CGA-e |
|---|---|---|---|---|---|---|---|
| **MNIST** | | | | | | | |
| Partition1 | 97.9 | 97.6 | 97.3 | 97.5 | 96.8 | 98.0 | 99.0 |
| Partition2 | 95.2 | 96.7 | 98.8 | 96.8 | 95.4 | 98.4 | 98.9 |
| Partition3 | 98.3 | 98.6 | 98.0 | 96.6 | 97.6 | 98.4 | 99.1 |
| Partition4 | 98.4 | 98.6 | 98.3 | 97.9 | 98.4 | 97.6 | 98.7 |
| Partition5 | 98.0 | 98.8 | 99.1 | 96.5 | 97.7 | 98.9 | 99.3 |
| Range | 3.2 | 2.1 | 1.8 | 1.4 | 3.0 | 1.3 | 0.6 |
| **CIFAR-10** | | | | | | | |
| Partition1 | 62.4 | 62.2 | 64.2 | 77.3 | 81.8 | 81.6 | 79.7 |
| Partition2 | 61.8 | 62.2 | 65.7 | 75.3 | 76.3 | 84.8 | 80.6 |
| Partition3 | 67.3 | 70.4 | 70.8 | 88.7 | 93.5 | 91.9 | 91.5 |
| Partition4 | 64.5 | 66.1 | 67.7 | 77.3 | 85.2 | 89.7 | 89.9 |
| Partition5 | 69.9 | 71.4 | 72.3 | 81.2 | 86.6 | 83.3 | 86.3 |
| Range | 8.1 | 9.2 | 8.1 | 13.4 | 17.2 | 10.3 | 11.8 |
| **SVHN** | | | | | | | |
| Partition1 | 88.4 | 89.3 | 89.9 | 92.7 | 93.3 | 91.3 | 90.5 |
| Partition2 | 90.0 | 90.4 | 91.7 | 92.1 | 91.9 | 90.4 | 91.3 |
| Partition3 | 90.8 | 90.3 | 92.1 | 94.2 | 95.3 | 92.1 | 92.3 |
| Partition4 | 90.8 | 91.4 | 92.4 | 95.2 | 94.6 | 92.7 | 93.2 |
| Partition5 | 91.6 | 91.8 | 92.5 | 95.2 | 94.3 | 92.5 | 93.3 |
| Range | 3.3 | 2.4 | 2.6 | 3.2 | 3.4 | 2.3 | 2.8 |
| **TinyImageNet** | | | | | | | |
| Partition1 | 57.2 | 57.9 | 58.9 | \ | \ | 73.2 | 75.0 |
| Partition2 | 56.4 | 56.3 | 56.9 | \ | \ | 73.0 | 74.0 |
| Partition3 | 61.9 | 62.3 | 62.9 | \ | \ | 71.2 | 73.0 |
| Partition4 | 60.3 | 60.3 | 60.6 | \ | \ | 74.4 | 75.8 |
| Partition5 | 51.6 | 52.8 | 51.2 | \ | \ | 69.3 | 70.8 |
| Range | 10.3 | 9.5 | 11.7 | \ | \ | 5.1 | 5.0 |

## B    DETAILED EXPERIMENT RESULTS

We found that when conducting OOD detection experiments using the same dataset (i.e., some classes of the dataset are used as the IND classes and the rest as the OOD classes), the choices of IND and OOD classes can affect the results greatly. Some partitions of IND and OOD classes are very hard while some are easy. Therefore, we fix 5 sets of IND and OOD classes to make the comparison between different methods fair. Table 4 shows the average AUC score for each of the 5 partitions of each dataset. From Table 4, we learn that the ranges (or differences) of the results for different IND and OOD partitions of the datasets are very large. On the CIFAR-10 dataset, the difference is as large as 17.2%.

## C    VISUALIZATION OF DATA DISTRIBUTION

We visualize the latent representations of different data to learn their distributional property. The visualization is done on partition 1 of MNIST dataset. Figure 4 shows the distribution of the generated data after dimensionality reduction by t-distributed Stochastic Neighbor Embedding (t-SNE) method. The left figure shows 6 clusters of IND data of 6 classes. Red points in the middle figure represent our generated pseudo data. This figure demonstrates that a large number of pseudo samples are around the boundaries of the clusters (representing the known classes). These are the most effective OOD-like samples. At the same time, our generated data distribute widely, which shows that the pseudo data is of great diversity. Pink points in the right figure represent real OOD data during testing. Comparing red points and pink points, we can learn that the diverse pseudo data can cover nearly all areas where there are true OOD instances.

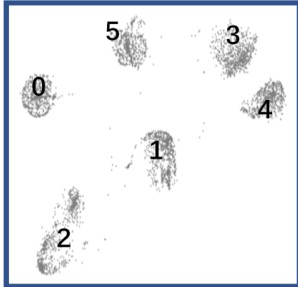 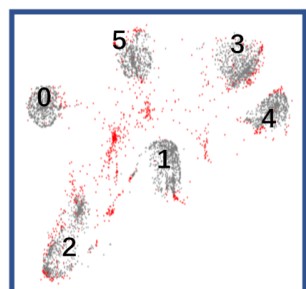 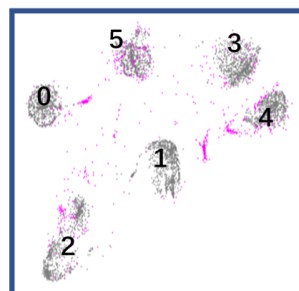

Figure 4: Distribution of IND data and the generated pseudo OOD data. The grey points represent the IND data and the numbers are their class labels. The red points in the middle figure represent the generated pseudo OOD data and the pink points in the right figure represent the true OOD data used in testing. The left figure contains only the IND data of 6 classes.

## D    VISUALIZATION OF PSEUDO OOD IMAGES

Figure 5 shows the generated pseudo OOD images from our CVAE. For each group, we first employ the standard class embedding and sample latent variables to generate 2 IND images. Then we use these 2 class embeddings to compute our pseudo class embedding (see the paper) and then generate pseudo OOD images with variables or features from different sampling distributions $\mathcal{N}(\mathbf{0}, \sigma^2 * \mathbf{I})$. It is easy to see some characteristics of IND images from pseudo images. When $\sigma = 1$, pseudo OOD images still resemble the IND images a great deal. But as $\sigma$ grows, the difference between the IND images and pseudo OOD images gets greater and the diversity also increases. That also explains why changing $\sigma$ can improve the final performance.

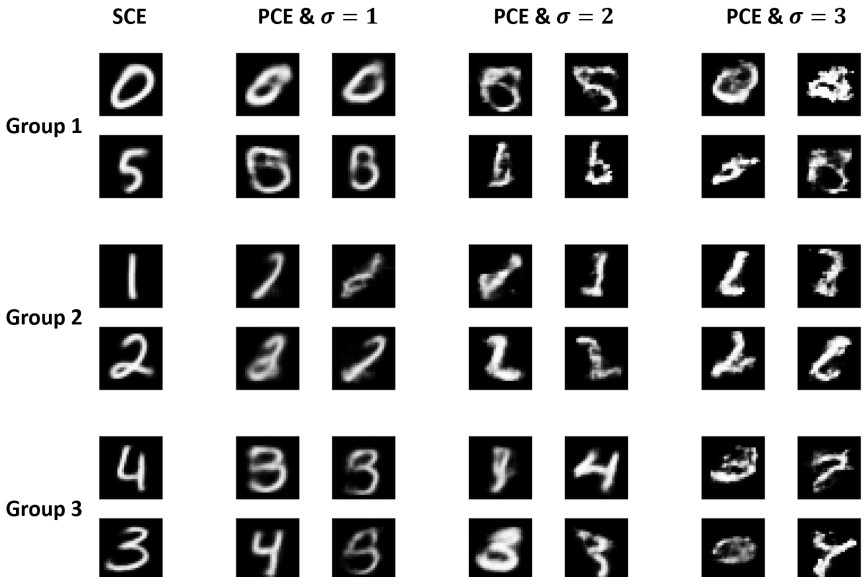

Figure 5: Generated data of different settings. SCE means *standard class embedding* and this column shows the generated data from CVAE using the standard class embedding. PCE means *pseudo class embedding* computed using the class embeddings of the two classes in each group (e.g., 0 and 5 in Group 1). For each setting of PCE & $\sigma$, we generate 4 images by sampling 4 different $z$ values. $\sigma$ is the parameter of the sampling distribution for $z$, $\mathcal{N}(\mathbf{0}, \sigma^2 * \mathbf{I})$.

