# OpenReview forum: "Efficient Out-of-Distribution Detection via CVAE data Generation"
_ICLR.cc/2022/Conference — ICLR 2022 Submitted_

### Official Review · Reviewer_eP8b · 2021-10-30

**Correctness:** 3
**Technical Novelty And Significance:** 3
**Empirical Novelty And Significance:** 3
**Recommendation:** 6
**Confidence:** 4

**Main Review:**

Strengths
1) Use of the class-conditioned generative models to generate pseudo-OODs seems novel and interesting to me.
2) The paper is well-organized and easy to read.
3) Result of the paper is competitive against the SOTA method, CSI, while the paper benefits from a 10x decrease in computation time.

Weakness
1) The proposed strategy requires knowledge of the labels, while CSI can be operated without labels via utilizing data augmentation. Following the analogy, I think adding more baselines that use knowledge from the classifiers can be beneficial. (e.g. [1])

Comments.
1) As mentioned in the previous section, I suggest adding more baselines that perform OOD detection from pre-trained classifiers.
2) I'm curious to see the ablation study that does not freeze the feature extractor in the 2nd phase of training CGA.
3) As CGA utilizes label information for training the CVAE, I am curious about the robustness of CGA in the case of corrupted, or noisy labels. For example, when no labels are given, we can generate the pseudo-label from clustering and apply the labels to the pipeline.
4) How's the result in the CIFAR-100 dataset? while I see the results on the tinyImageNet dataset, I think the result would be strengthened by adding CIFAR-100 results.


***********References**********
[1] Detecting Out-of-Distribution Examples with Gram Matrices

**Summary Of The Paper:**

UPDATE2

I acknowledge that I've read the author responses as well as the other reviews

I think the authors gave a good explanation of where their proposed method outperforms the conventional methods. Therefore, I increase the score to 6 weak accept.

=========================================================================

UPDATE:
I acknowledge that I‘ve read the author responses as well as the other reviews.
I initially gave 6 weak accept to the paper. However, after reading the author response and other reviewers' comments, I think the proposed work lose its merit due to several reasons.

1. The proposed model utilizes label information and underperforms against conventional methods that utilize label information. (see the comments). As such methods do not train an extra model, the proposed method's efficiency becomes also vague.

2. As Reviewer BUDw mentioned, I also agree that CVAE possibly becomes inefficient when the image size becomes larger.

3. Furthermore, as the performance and efficiency become vague, I'm not sure about the novelty, or how "surprising" the proposed method can be. The application of the generative models to OOD detection has a long history and better generative models will give better results.

Due to the following reasons, I am leaning towards rejection of the paper and downgrading the score to 5. I hope the authors clarify the issues.

================================================================================================


The paper proposes CGA (CVAE-based generative data augmentation for out-of-distribution (OOD) detection) to improve the OOD detection performance when the class information is given. The module consists of a classification module (classifier and feature extractor) and a generation module (encoder and decoder). In the first phase, only in-distribution data are used to train two modules. In the second phase, CGA utilizes the generated pseudo-OOD data to fine-tune the classifier. CGA is compared against SOTA baselines in various setups. In the majority of setups, CGA shows competitive performance against the conventional OOD detection methods with the advantage of reduced computation times. Various ablation supports the design of CGA.

**Summary Of The Review:**

Overall, I liked the proposed algorithm and think that CGA can be a competitive baseline for supervised OOD detection. Therefore I'm leaning towards accepting the paper.

---

### Official Review · Reviewer_N6Uq · 2021-10-30

**Correctness:** 4
**Technical Novelty And Significance:** 3
**Empirical Novelty And Significance:** 3
**Recommendation:** 6
**Confidence:** 4

**Main Review:**

The paper is well written, covers the related works, and importantly, is well motivated.
The usage of 'known OOD' samples in order to improve OOD detection is very common, and while in some times there is abundance of such `known OOD` data, that is often not the case.

The proposed method is rather simple, and is implemented in two baselines, creating surprisingly good results.

The reason the results are surprising, in the CIFAR-10 experiments (in my opinion),
is that extrapolating on the latent space will often results with samples that in high probability will not resemble any thing the model will see in the future (OOD or ID).
In addition, the results of a VAE reconstruction are often blurry, unless explicit efforts has been made to prevent that, which none were done.

To summarize, I would expect the model to learn how to classify the generated output of the VAE as OOD, while 'real' images, such as CIFAR-100 (which was used as OOD set) still get classified ad inset.

If the authors could elaborate - what was done to prevent the above?

An interesting ablation would be to use `Known OOD` data in the same baselines, to measure how well the generated samples are compared to real OOD data.

In addition, results on more difficult datasets could contribute - Cifar100 as IND, or even the entire TinyImageNet as IND. The current partitioning of TinyImageNet leaves only 20 classes IND in each partition, which is not 'difficult' enough.

**Summary Of The Paper:**

The authors proposed to use a Conditional VAE in order to generate pseudo OOD data, which can be used to improve OOD detection.
The authored demonstrated the method in several benchmarks, with good results.

**Summary Of The Review:**

The authors proposed what seems like a simple solution (which is great in my opinion) for an important problem, however they have overcome several challenging problems and achieved good results.
The paper itself is well written, easy to follow, and is very detailed regarding the implementation and experiment settings.

====================================

Following the discussion with the other reviewers, I have decided to keep my current score. As other reviewers have pointed out, the selection of the CVAE is not justified enough, and a serious ablation study should be added to justify it and not other generative methods.

---

### Official Review · Reviewer_BUDw · 2021-11-01

**Correctness:** 3
**Technical Novelty And Significance:** 2
**Empirical Novelty And Significance:** 2
**Recommendation:** 5
**Confidence:** 4

**Main Review:**

Strengths:

* I'd say the main strength in this paper is the empirical results - particularly the combination with CSI seems to help, sometimes significantly.
* In the same vain as the above strength, the hyper-parameter ablations are quite thorough - and seem to cover the hyperparameters that I find interesting.
* The paper is quite clear and easy to follow.

Weaknesses:

* Novelty: using generative models has been extensively done in anomaly detection in many ways including synthesis of anomalies (e.g. [1]). The basic ideas are similar to this paper (but as there has been a lot of previous work, there are also some more nuanced ideas).
* CVAE are only fast for small images - larger images are very hard to generate reliably, particularly with VAE. Conveniently, the datasets evaluated here are small-scale but I find it hard to imagine the sa,e results would follow for ImageNet or another suitable large-scale dataset.
* The choice of CVAE does not seem to have been seriously ablated. There are many CVAEs possible as well as GAN-based and more recent models. This comparison could have significantly helped the significance of this work.
* The advantage of training time is interesting, but this (and better accuracy) can easily be achieved by using ImageNet-pretrained representations. E.g. [2] and earlier references. Although these methods do not apply for tabular data, but I suspect that CVAE will also not work well there.

[1] Pourreza et al., G2D: Generate to Detect Anomaly, WACV'21
[2] Fort et al.,  "Exploring the Limits of Out-of-Distribution Detection, NeurIPS'21.


**Summary Of The Paper:**

This paper suggests to model outliers using a CVAE. It then suggests simple scoring functions for incorporating these simulated outlier into a classifier and uses a confidence based criterion. A numerical comparison shows this method is able to outperform CSI on a couple of datasets (while it underperforms on most other ones) . It also shows that synthetic anomalies can be combined with existing methods and obtain gains in performance.

**Summary Of The Review:**

Although the empirical results are encouraging, I believe this paper still does not make the bar in two aspects: i) conceptually - the ideas of synthesizing anomalies or using generative models for this purpose are not new, and the techniques used here are not new either - not has there been a very extensive analysis of them ii) practically - better results can be obtained using other methods such as ImageNet-pretraining, so in terms of practical utility to the engineer - this might not be the first choice.

############################

I thank the authors for the response. I do not believe it addresses my main concern i.e. lack of convincing ablation. I therefore kept my borderline reject rating.

---

### Official Review · Reviewer_RyWJ · 2021-11-02

**Correctness:** 3
**Technical Novelty And Significance:** 3
**Empirical Novelty And Significance:** 3
**Recommendation:** 5
**Confidence:** 3

**Main Review:**

Strengths
1.This paper claims that they propose a general and much more efficient out-of-distribution detection model.
2.This paper shows that a simple CVAE model can generate pseudo OOD data to assist the training phase.

Weaknesses
1.It is obvious that this paper applies CVAE to the OOD data detection. The question is why to select CVAE as the efficient model to generate the OOD data. What is the motivation?
2.This paper claims that we can already produce comparable results to existing SOTA contrastive learning models but much more efficient. Why? The detailed explanation is necessary.
3.The contribution is mainly the metrics.

**Summary Of The Paper:**

This paper proposes to employ a CVAE structure to generate pseudo OOD samples by providing some synthetic conditional information. Besides, they design a two-stage framework to train an OOD detection model by leveraging the generated pseudo OOD data. They show that their approach outperforms other SOTA methods in the task of out-of-distribution detection.

**Summary Of The Review:**

Overall, the paper does not have enough interesting results for acceptance.

---

### Decision · Program_Chairs · 2022-01-20

**Decision:**

Reject

**Comment:**

The paper adopts CVAE to generate OOD samples for training an outliner detector. It consists of two phases that train an OOD detector by leveraging the generated OOD data and shows it outperform other methods. According to reviewers’ discussion, there is a concern from the discussion: why CVAE works but other variants or cGAN doesn’t. The paper needs more motivation or evidence or ablations to support the generality of the work.